# Endovascular Abdominal Aortic Aneurysm Repair: Overview of Current Guidance, Strategies, and New Technologies, Perspectives from the United Kingdom

**DOI:** 10.3390/jcm11185415

**Published:** 2022-09-15

**Authors:** Robert Bakewell, Miltiadis Krokidis, Andrew Winterbottom

**Affiliations:** 1Department of Radiology, Cambridge University Hospitals NHS Foundation Trust, Cambridge CB2 0QQ, UK; 21st Department of Radiology, National and Kapodistrian University of Athens, Areteion Hospital, 15772 Athens, Greece

**Keywords:** abdominal aortic aneurysm, endovascular repair, NICE guidelines, FEVAR, BEVAR, endoachors, chimneys

## Abstract

Endovascular aortic aneurysm repair has changed the management of patients affected by this condition, offering a minimally invasive solution with satisfactory outcomes. Constant evolution of this technology has expanded the use of endovascular devices to more complex cases. The purpose of this review article is to describe the current strategies, guidance, and technologies in this field, with a particular focus on practices in the United Kingdom.

## 1. Introduction

Abdominal aortic aneurysm (AAA), defined as an abdominal aorta exceeding the normal vessel diameter by over 50%, affects approximately 4% of the male population in the UK [1]. The overwhelming majority of cases are secondary to atherosclerosis, with risk factors including smoking, hypertension, age and male sex [2].

The natural history of AAA involves the growth of the aneurysm and rupture, with the probability of both increasing with aneurysm size. The mortality rate from ruptured AAA is >50% and, therefore, detecting and treating AAAs before this occurs is key to reducing AAA-related morbidity and mortality. For aneurysms between 5.5 cm and 6.9 cm, the annual rupture risk is approximately 9%, whereas this increases to >32% when AAAs expand beyond 7 cm [3]. Although age-standardised death rates from AAA have reduced in Europe and the USA over the past 30 years, AAA still accounts for 5 and 2 deaths per 100,000 in men and women, respectively [4]. The reason for the declining death rates is multifactorial, with contributions from behavioural changes within the population, as well as earlier detection and treatment. Tobacco smoking, a major modifiable risk factor, has significantly reduced in Europe and the USA, which is also reflected in the declining incidence of other atherosclerotic diseases, such as myocardial infarction and peripheral arterial disease. In addition, the UK, Sweden and the USA have national AAA screening programs, which have been shown to reduce AAA-related mortality [5].

The mainstays of management for AAA are open surgical repair (OSR) and endovascular aneurysm repair (EVAR). Although OSR is a major abdominal surgery, with the early operative risks that come with it, it is seen as a definitive fix and does not require routine follow-up.

EVAR is an endovascular technique where stent grafts are deployed from inside the aorta through a distant (typically common femoral) artery access to exclude the aneurysm (Figure 1 and Figure 2). While modern technologies push the boundaries of what is possible with EVAR, there are still strict criteria in the manufacturers’ instructions for use (IFU) for anatomy regarding what is and is not suitable for EVAR. This mostly relates to ensuring the stent graft is sealed effectively to be able to exclude the aneurysm from the circulation. Unlike OSR, patients undergoing EVAR require regular clinical and imaging follow-up to monitor for complications such ‘endoleak’, whereby blood flows into the supposedly excluded aneurysm sac, placing it under tension and risking growth and rupture. This can occur through several mechanisms [6]. As aneurysmal disease is typically progressive, what was once an adequate sealing zone may, over time, dilate and distort to such an extent that the EVAR seal is no longer satisfactory. The post-EVAR aneurysm sac may also fill with blood that flows in a retrograde manner from the aortic branches (typically the inferior mesenteric artery or lumbar arteries), or from loss of the seal of the graft components to each other. Depending on the type of endoleak and various other aneurysm and patient factors, revision endovascular procedures can be performed in an attempt to prevent sac rupture.

The risk of rupture post-EVAR is related to the aneurysm sac size and growth, and hence the major goal of EVAR is to exclude and shrink the sac [7,8]. Since the turn of the century, there has been a general change in practice worldwide towards EVAR over open surgical repair. For example, in the USA only 5.2% of patients underwent EVAR for elective AAA in 2000, compared to 74% in 2010 [9]. In recent years, there has also been an increase in the endovascular management of AAAs previously considered unsuitable for EVAR. These involve more advanced endovascular techniques, such as fenestrated and branched EVAR (FEVAR/BEVAR), chimney/snorkel devices, and endoanchors. These will be discussed in more detail later.

## 2. EVAR vs. OSR in Unruptured Infrarenal AAA

Despite the global shift towards AAA management with EVAR over OSR in the past 20 years, in recent years there has been much debate and controversy surrounding the optimal management of unruptured infrarenal AAA in the UK.

The accepted threshold for infrarenal AAA treatment in the UK (and worldwide) is 5.5 cm or >4 cm, but >1 cm/year growth [10]. At this size, the risk of aneurysm rupture during surveillance is higher than the periprocedural mortality from OSR and, therefore, the benefits of repair begin to outweigh the risks. Despite the fact that the original trials that informed this decision were performed over 20 years ago and only used OSR as the intervention, no study since has unequivocally proven that there is a mortality benefit to repairing unruptured aneurysms <5.5 cm, either with EVAR or OSR [11,12]. The recommendation for these smaller aneurysms is that they should undergo imaging surveillance and then be assessed for potential repair if and when the size/growth threshold is met.

Several randomised controlled trials have assessed the relative benefits of EVAR vs. OSR for unruptured infrarenal AAA, including EVAR-1, OVER, DREAM and ACE [13,14,15,16]. The consensus is that 30-day mortality rate is lower for EVAR than OSR [13]. However, the all-cause mortality benefits of EVAR reduce relative to OSR over time, with no difference between the two procedures at 3–5 years [17]. Indeed, long-term follow-up in the EVAR-1 trial has shown that by 8 years post-intervention, both aneurysm-related mortality and all-cause mortality are higher in patients treated with EVAR vs. OSR [18]. In addition, EVAR is associated with a higher reintervention rate and requires lifelong surveillance for endoleak, which adds to the cost, and likely reduces quality of life. The delayed AAA-related mortality rate seen in the multiple publications from the EVAR-1 trial, from which these figures are taken, is generally seen as a result of sac expansion and rupture secondary to endoleak. Therefore, current recommendations in the UK from the National Institute for Health and Care Excellence (NICE) are that patients who are fit enough to undergo OSR for unruptured AAA should do so, and EVAR should be reserved for the remaining patients as long as their anatomy is appropriate. However, there is controversy around this, as randomised controlled trial data from the EVAR-2 trial have shown a lack of benefit of EVAR in patients unfit for OSR, even relative to nonoperative management [19].

In any case, in 2020 EVAR still accounted for 60% of all elective AAA repairs in the UK [20], which was unchanged from the year prior. It must be noted that total numbers undergoing repair had reduced, and criteria for repair had been altered (only aneurysms >70 mm were advised for repair by the Vascular Society of Great Britain and British Society of Interventional Radiologists) due to the extreme pressures placed on the National Health Service by the COVID-19 pandemic. As the UK and the global community continue to emerge from the COVID-19 pandemic, time will tell if the NICE recommendations are reflected in the real-world data. In the majority of cases in Europe, the USA and Australia, EVAR continues to be recommended as the primary treatment for unruptured infrarenal AAA. In the USA, EVAR makes up the majority of repairs of unruptured infrarenal AAAs, accounting for >80% of cases [21]. Given the evidence presented above, this heavy skew towards EVAR, and so decline in the number of OSRs, has led to concerns around deskilling in this important procedure, which is still the gold standard for definitive aneurysm treatment. Despite increasing experience in fixing complications related to EVAR, it is inevitable that a small proportion, approximately 1%, will require open surgical conversion [22].

## 3. Complex AAA Repair Strategies and New Technologies

Although there is no universally agreed-upon definition, the term complex abdominal aortic aneurysm is being increasingly used to describe AAAs that are anatomically unsuitable for standard infrarenal repair (either OSR or EVAR). This, therefore, encompasses a heterogeneous group of aneurysmal diseases, ranging from short- or conical-necked infrarenal aneurysms to suprarenal abdominal aortic aneurysms. These aneurysms may, therefore, involve a varying number of visceral arteries arising from the abdominal aorta. The open surgical repair of these complex aneurysms is challenging and carries a higher mortality and morbidity rate than OSR for standard infrarenal AAAs, although the exact figures are difficult to determine due to the heterogeneity of the disease processes and a lack of standardised definitions. In order for an endovascular repair to be performed in these scenarios, creating an adequate seal to the aorta is essential. Endoanchors may be used in infrarenal AAAs with necks too short to seal a standard EVAR device. When visceral arteries are involved in the aneurysm, these must be involved in the repair. Placing stent grafts into the visceral arteries and running these between the outer wall of a standard EVAR stent graft and the aortic wall itself forms the basis of the chimney technique. In the increasingly popular FEVAR/BEVAR (fenestrated EVAR/branched EVAR) procedure, stent grafts with fenestrations or branches at precise locations to allow cannulation and stent grafting of the visceral arteries can also be created. This allows large segments of a diseased aorta to be excluded from the circulation. While device technology is continually improving and these techniques are becoming ever more popular, there is a lack of high-quality evidence across the board. The next section will review these endovascular techniques in more detail.

### 3.1. Chimney EVAR (ChEVAR)

The chimney EVAR (ChEVAR) technique of running stent grafts from visceral arteries parallel to the main body of the EVAR was originally used as a rescue technique to preserve flow in visceral arteries that had been accidentally covered in standard EVAR procedures. However, it has now emerged as a technique in its own right for the endovascular management of complex AAA (Figure 3). When compared to OSR for juxtarenal AAA and Crawford-type IV thoracoabdominal aneurysms, ChEVAR has a significantly lower 30-day mortality rate (2% vs. 11.9%) [23]. In patients with juxtarenal AAA who are deemed unsuitable for FEVAR, there is no significant difference between ChEVAR and OSR in terms of aortic-related mortality rate at five years [24]. However, in the same study, the overall mortality rate at five years was found to be higher in the ChEVAR group and there was also a higher aortic reintervention rate (35.4% in ChEVAR vs. 9.5% in OSR). The main concerns regarding this technique are the ability to form an adequate seal proximally and, therefore, prevent a type 1a endoleak around the chimney (a so-called ‘gutter leak’), as well as patency of the chimneyed visceral stent graft(s). The Pericles registry was created to provide multi-centre international data on outcomes in patients treated with chimney techniques for pararenal aortic aneurysms, with 517 patients treated in 13 centres across Europe and the USA [25]. Key points to come out of the primary publication from this study are a 7.9% early gutter leak, with spontaneous resolution in the majority of cases leading to an overall type 1a leak rate of 2.9%, and a primary stent/stent graft patency of 94%. A mean aneurysm sac reduction to 61.2% after 17.1 months is also significant, as the goal of any endovascular AAA repair is to shrink the aneurysm to reduce the risk of rupture. The Pericles registry has shown that the more chimneys placed, the higher the risk of occlusion and mortality [26]. Therefore, it is currently recommended that as few chimneys as possible be placed (ideally one, and maximum two), that stent grafts be used (rather than bare metal stents), and that the ideal sealing zone should not be less than 20 mm. In line with this, the European Society of Vascular Surgery has recommended the use of these parallel graft techniques for juxtarenal aortic aneurysms, either in an emergency situation or when a fenestrated device is contraindicated, ideally using no more than two chimneys [27]. The benefit of this technique over other advanced endovascular repairs for complex AAA is the use of off-the-shelf devices, meaning that they can be performed in an emergency/semi-elective setting, as opposed to the custom grafts that are typically required for FEVAR/BEVAR. Chimney EVAR is also quicker to perform than FEVAR/BEVAR, which is a key benefit in emergency settings. Additionally, fenestrated and branched devices typically require larger-calibre iliac arteries for access than the low-profile devices that can be used in ChEVAR.

### 3.2. Fenestrated/Branched EVAR (FEVAR/BEVAR)

Fenestrated/branched EVAR is an increasingly popular method for treating complex abdominal aortic aneurysms, with a more than six-fold increase in these procedures in the USA in the first two years since its introduction in 2011 [28]. This has been mirrored in Europe, with a 24% increase in FEVAR/BEVAR in UK vascular units between 2014 and 2020 [20,29]. These typically custom-made aortic stent grafts can, with some anatomical limitations, include any number of the major visceral arteries, and can also be combined with thoracic endovascular stent grafts to manage aneurysms that also involve the descending thoracic aorta (Figure 4).

It must be highlighted that FEVAR/BEVAR in itself is also a very broad term, with multiple combinations of fenestrations, branches, scallops and additional thoracic and iliac extensions possible. Therefore, attempting to give accurate data on outcomes of ‘FEVAR/BEVAR’ as a whole and comparing these with other techniques is problematic and likely to be misleading. In addition, there are variations in the type of fenestrated devices that may be used. For example, although most fenestrated stent grafts are custom-made to match the visceral artery origins precisely, there are off-the-shelf alternatives that may be used in acute and semi-elective settings to overcome manufacturing times. These stent grafts have standardised fenestration locations, which have been shown to be anatomically suitable in up to 70% of juxtarenal aneurysms [30]. It is also possible for endovascular specialists to create their own fenestrations in an off-the-shelf stent graft at the time of the procedure. These physician-modified endografts (PMEG), in appropriately skilled hands, have been shown to be safe, but there is a lack of standardisation of the technique and limited data comparing their use with other endovascular techniques or OSR [31].

As with all advanced endovascular techniques for complex AAA, there are a lack of high-quality data comparing FEVAR/BEVAR with open surgical repair or with other endovascular procedures. Two of the three systematic reviews that have assessed OSR vs. FEVAR for juxtarenal AAA have found a lower early postoperative mortality rate in those patients treated endovascularly (0.9–1.4% vs. 2.5–3.6% for FEVAR and OSR, respectively) [32,33]. In the third systematic review, early mortality rates were equal between the two procedures at 4.1% [34]. A systematic review and network meta-analysis that compared multiple endovascular techniques with OSR for the treatment of short-necked and juxtarenal AAA found a six-fold reduction in perioperative mortality with FEVAR vs. OSR [35]. All-cause mortality rates, however, were equal between the two procedures by 2.5 years. This study also highlighted the lack of high-quality evidence in this field, with no randomised trials available. A key finding across multiple studies is the high reintervention rate post-FEVAR/BEVAR, which is usually as a result of endoleak. This, along with the requirement for regular clinical and radiological follow-up, ultimately adds to the cost of the procedure. High visceral stent graft patency (>95% at 1 year) has been shown across multiple studies [36,37]. A long-term follow-up to 12 years has been published, which confirms the safety of the FEVAR procedure [38]. In this study, while the 8-year survival rate was only 20%, only 2% of deaths were aneurysm-related. This reflects the population in question, which is typically of advanced age with multiple comorbidities, and hence the attraction of a less invasive method of fixation.

Comparing FEVAR with other endovascular techniques is challenging in this highly heterogeneous population, which creates problems matching patients. However, a systematic review and meta-analysis have shown a lower aneurysm-related mortality rate in patients treated with FEVAR vs. ChEVAR (1.4% vs. 3.2%), at the cost of a higher reintervention rate (11.7% vs. 5.6%) [39]. The SOCRATES short-necked aneurysm randomised trial, which is currently recruiting, aims to compare FEVAR with endoanchors in short-necked (4–15 mm) abdominal aortic aneurysms. This study, with an estimated completion in 2026, should provide some high-level evidence for endovascular treatment in this population.

While FEVAR/BEVAR is less invasive than open surgical repair for complex AAA, it is still physiologically demanding for the patient, as well as technically demanding for the operator. Patent selection is therefore crucial, both from an anatomical and physiological perspective, and appropriate pre-procedure work-up and optimisation is essential [40].

### 3.3. Endoanchors

Endoanchors are endovascularly inserted screws that attach a proximal stent graft to the aortic wall. They can be inserted either during the primary procedure, in cases where the aneurysm neck would be unfavourable for standard EVAR, or as a secondary procedure to treat a type 1a endoleak (blood leaking into the aneurysm sac past the proximal seal zone) (Figure 5). With the use of this technology, infrarenal aneurysms with necks as short as 4 mm (Helix-FX, Medtronic) can be treated with EVAR, whereas the shortest neck that can be safely treated with a standard repair is generally accepted to be 10 mm (Endurant II/IIs, Medtronic). While early studies with endoanchors are promising, there is a lack of high-quality evidence, with most studies being observational, and a lack of case-controlled trials. The ANCHOR registry is an industry-led prospective observational study following patients with endoanchors (Heli-FX EndoAnchor, Medtronic) inserted in multiple centres in Europe and the USA [41]. Four-year data have been presented, which show these devices are safe and effective, with a type 1a endoleak rate of 3.4%. There were two systematic reviews published in 2020 assessing studies involving endoanchors [42,43]. Both systematic reviews have commented on the lack of long-term follow-up, but type 1a endoleak rates of 3.5% and 6.2% have been reported for primary endoanchor interventions. It is worth noting that within the studies included in these reviews, the mean AAA neck characteristics would not meet the criteria to be labelled as ‘hostile’, which is a major indication for the use of primary endoanchors. Therefore, more data are needed on efficacy in the patient cohort in which these devices will be used, which will hopefully come from the ANCHOR registry in due course. In the UK, NICE advises that any endoanchor use (primary or secondary) should be discussed with the patient regarding the lack of data, with special arrangements for clinical governance, consent, audit and research [44].

### 3.4. Endovascular Repair of Aortoiliac Aneurysms

With improvement in operator experience and device design, aortic aneurysms that extend into the iliac arteries may also be treated endovascularly. The advent of the iliac branch device (IBD) has allowed the treatment of these aneurysms with preservation of the flow into the internal iliac artery (Figure 6). This negates the risk of buttock claudication, erectile dysfunction and bowel ischaemia, which has been seen with previous endovascular techniques that required occlusion of this artery.

## 4. Conclusions

Endovascular repair of abdominal aortic aneurysms is becoming feasible for ever more complicated aneurysm morphology, thanks to a combination of improved technology and increased operator experience. With the rapid adoption of new technologies and techniques, it is vital that high-level evidence is acquired and that outcomes are rigorously scrutinised to ensure patients receive the most appropriate care.

## Figures and Tables

**Figure 1 jcm-11-05415-f001:**
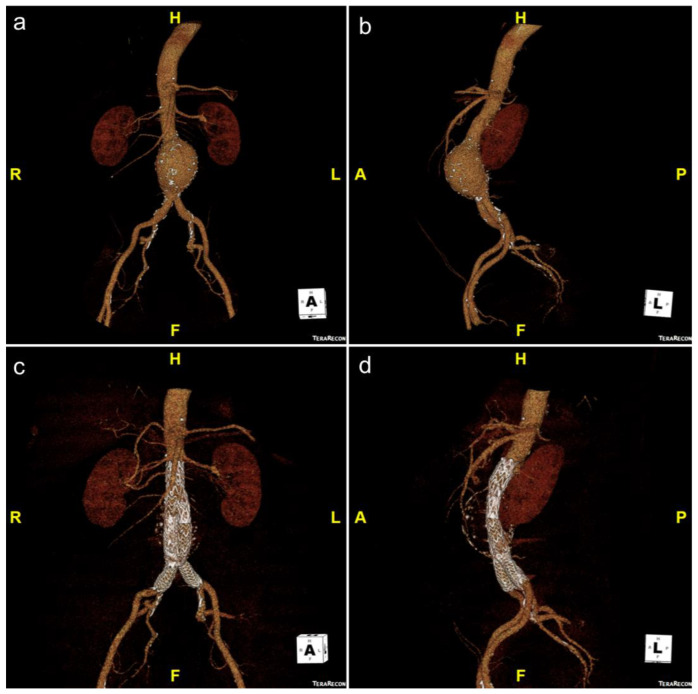
Volume-rendered CT angiogram showing an infrarenal abdominal aortic aneurysm in frontal (**a**) and lateral (**b**) orientations. The patient underwent elective EVAR, which is demonstrated in (**c**,**d**).

**Figure 2 jcm-11-05415-f002:**
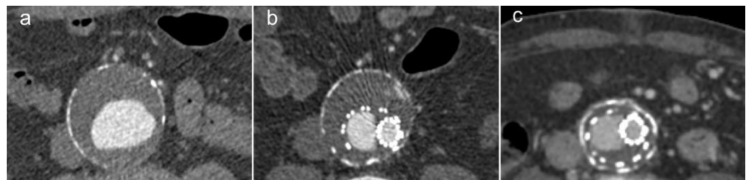
(**a**) CT angiogram showing an infrarenal abdominal aortic aneurysm. The patient underwent EVAR with follow-up CT 1 year (**b**) and 16 years (**c**) post-procedure. Note the reduction in sac size over time.

**Figure 3 jcm-11-05415-f003:**
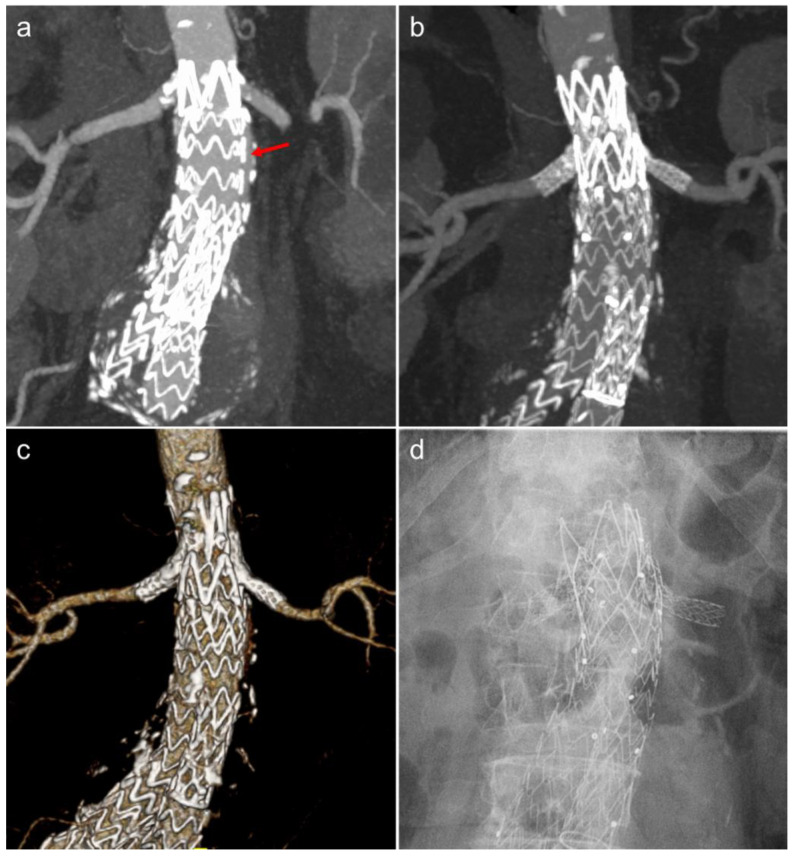
(**a**) AP coronal maximum-intensity projection (MIP) image of a patient with a type 1a endoleak (red arrow) post-EVAR. The patient underwent chimney stent grafts to both renal arteries with an aortic cuff extending above the renal arteries to create a new proximal neck seal. Coronal MIP (**b**), volume-rendered CT (**c**), and abdominal radiograph (**d**) showing the chimney EVAR.

**Figure 4 jcm-11-05415-f004:**
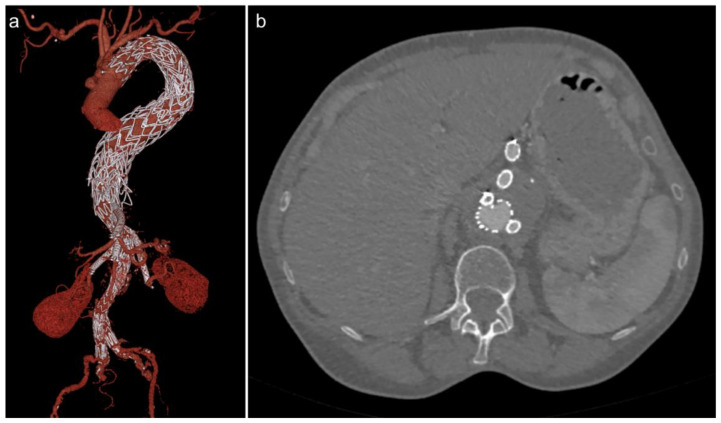
(**a**) Volume-rendered image of a CT angiogram post-FEVAR with thoracic stent grafts for a thoracoabdominal aneurysm. (**b**) Axial slice of the same CT angiogram showing stent grafts in four aortic branches (coeliac, superior mesenteric, and bilateral renal arteries).

**Figure 5 jcm-11-05415-f005:**
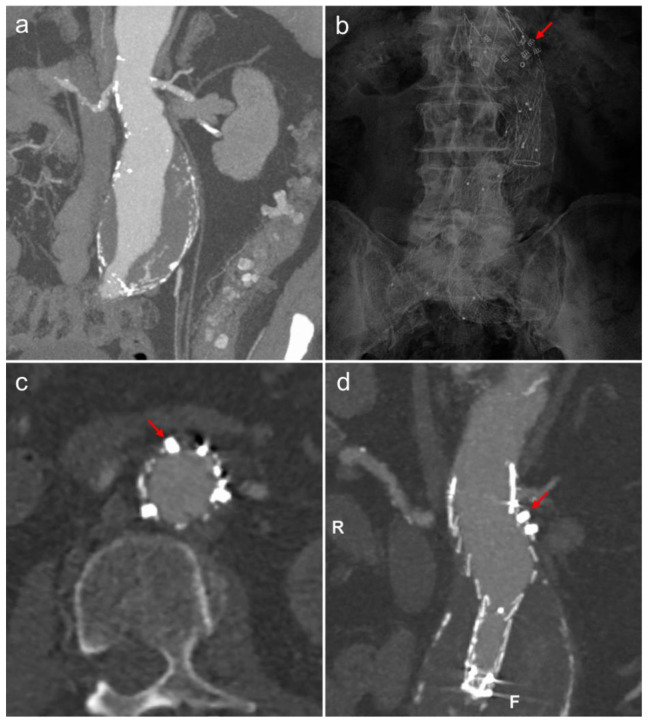
(**a**) Coronal oblique MIP image of an abdominal aortic aneurysm with a conical neck unsuitable for conventional EVAR. The patient was treated with EVAR with primary endoanchors (Heli-FX, Medtronic). Follow-up abdominal radiograph (**b**), axial (**c**) and coronal (**d**) CT angiogram demonstrating the endoanchors attaching the proximal stent graft to the aneurysm neck (red arrows).

**Figure 6 jcm-11-05415-f006:**
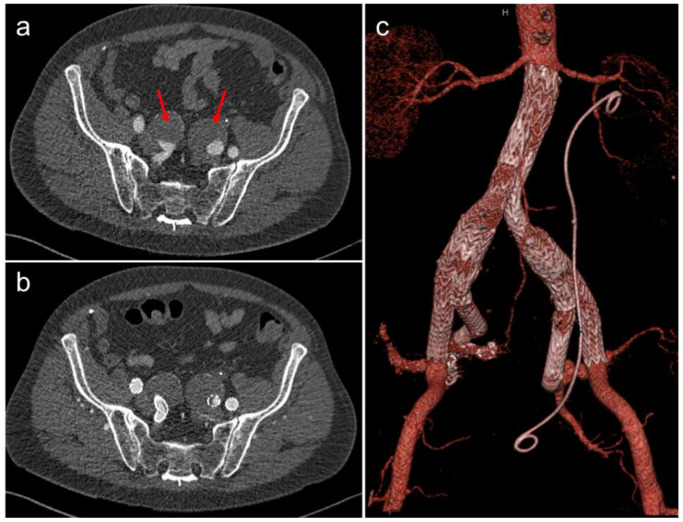
(**a**) CT angiogram showing bilateral internal iliac artery aneurysms (red arrows). The patient underwent EVAR with bilateral iliac branch devices (IBD). (**b**) Follow-up CT post-repair. (**c**) Volume-rendered image of the same post-operative CT showing the EVAR and bilateral IBD. Note the incidental left ureteric stent.

## Data Availability

Not applicable.

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
