# Peer review of "Endovascular Abdominal Aortic Aneurysm Repair: Overview of Current Guidance, Strategies, and New Technologies, Perspectives from the United Kingdom"

_jcm, 2022, doi:10.3390/jcm11185415_

Round 1
Reviewer 1 Report
Dear authors, it has been a pleasure to review your manuscript. Here are some considerations:
Abstract
Introduction
- Perhaps the definition of “exceeding the normal vessel diameter by > 50%” would be more correct.
- There are so many abbreviations without their significant (for example MI or PDV ect), you should write also the full name.
- Open repair is the milestone because it treats the aneurysm, endo leaves the aneurysm there and is in my opinion always an alternative solution.
- From several studies we have seen that the overall mortality after elective EVAR is higher than that of OSR .. maybe we will have a change in our clinical practice with a return to OSR?
- I suggest delete the last 4 lines of the introduction.
Complex AAA repair strategies and new technologies
- I think complex AAA cannot include type IV of TAAA or branched EVAR because the disease is too different.
- Consider to move the Chmney paragraph before the endoanchrs (they are a newer technique).
- I suggest you to mention “Menegolo M, Xodo A, Penzo M, Piazza M, Squizzato F, Colacchio EC, Grego F, Antonello M. Open repair versus EVAR with parallel grafts in patients with aneurysm of the abdominal aortic juxtarenal excluded from fenestrated endotransplant . J Cardiovasc Surg (Turin). 2021 Oct; 62 (5): 483-495. doi: 10.23736 / S0021-9509.21.11833-6. Epub 2021 Jun 18. PMID: 34142524. In this article they have seen that OSR is probably still the best option for JAAA excluded from FEVAR, although no differences in aortic-related mortality were observed.
What to say about the complications in the treatment of these conditions and their management? What about the experience of the centers? It is easy to deploy a standard EVAR and this requires a lower learning curve than OSR, but will there be surgeons able to do OSR if the trend is not reversed? What about the incidence of open conversion after EVAR and its peculiar aspects?
I suggest you read and quote this work "Xodo A, D'Oria M, Mendes B, Bertoglio L, Mani K, Gargiulo M, Budtz-Lilly J, Antonello M, Veraldi GF, Pilon F, Milite D, Calvagna C, Griselli F, Taglialavoro J, Bassini S, Wanhainen A, Lindstrom D, Gallitto E, Mezzetto L, Mastrorilli D, Lepidi S, DeMartino R. Peri-Operative Management of Patients Undergoing Fenestrated-Branched Endovascular Repair for Juxtarenal, Pararenal and Thoracoabdominal Aortic Aneurysms: Preventing, Recognizing and Treating Complications to Improve Clinical Outcomes. J Pers Med. 2022 Jun 21; 12 (7): 1018. Doi: 10.3390 / jpm12071018. PMID: 35887518; PMCID: PMC9317732. "
There is a lack of references on the use of fenestrations in situ and physician modified endographs, which are especially important in urgent cases.
Reviewer 2 Report
Reviewer comments:
The authors present an overview of currents strategies for endovascular abdominal aortic aneurysm repair focusing on the clinical practice in UK.
Overall the authors performed a nice synopsis of the available stentsgraft configurations and strategies.
There are, however, some major points to be addressed:
1. The manuscript focus on the clinical regulation and guidelines in UK, this should be cited in the abstract and title.
2. The presentation require more detailed statements with the corresponding literature reference. For example line 96 “several randomized controlled trials have assessed” would be implemented in “Trial A, B, C, D have assessed..” (see detailed comments below).
3. In the Conclusion (line 303-305) the authors seems to sustain, there is “in general” not enough evidence for EVAR/OSR vs no intervention. This can be quite misleading for the readers, leading to the concept that AAA eventually require no treatment at all. Please clarify.
Detailed comments
Line 20-24: Please provide the reference of these data
Line 31-32 “ and there is much pre-clinical work assessing novel biomarkers for AAA growth and rupture.” Please provide the reference
Line 40 “MI” “PVD”: these abbreviations need to be explained
Line 40- 42: “ In addition, the UK, Sweden and the USA have national AAA screening programs, which has been shown to reduce AAA related mor tality.”Please provide the reference of these data
Line 54-56 “Unlike OSR, patients undergoing EVAR require regular clinical and imaging follow-up to monitor for 53 complications such 'endoleak', whereby blood flows into the supposedly excluded aneurysm sac, placing it under tension and risking growth and rupture. This can occur through several mechanisms.” Please define these endoleak or report to a previously published open access figure (For example Figure 1, doi 10.3390/biomedicines10061442)
Line 96 “several randomized controlled trials have assessed (…)” please specify the trails your refer
Line 112-126: These description of the U-turn in the UK regulation could be reduced or removed, considering this is of limited impact for the readers
Line 152, line 170-173: the topic of challenging neck have been quite extensively discuss and recently sistematically reviewed (for example Eur J Vasc Endovasc Surg 2022 May;63(5):696-706. doi: 10.1016/j.ejvs.2021.12.042.).
Line 231 “there are some anatomical contraindications to FEVAR/BEVAR, which do not preclude the use of the chimney technique.” Please cite the anatomical characteristics which represent a contraindication for FEVAR/BEVAR but not for chimney
Line 241 The author presents FEVAR/BEVAR as custom-made aortic stentgrafts in opposite to chEVAR, as off-the shelf solution. What about “T-branch” as off-the shelf solution? Why was this not reported in the overview?
Overall the authors performed a pleasant overview of the available technologies with clear figures. Whith proper major revisions, this paper could be of interest for the readers outside the endovascular field willing to get better understanding of the EVAR therapeutic spectrum. However a more detailed references-work-up and a clearer statement in the conclusion are strongly suggested.
Round 2
Reviewer 1 Report
I have no other observations or comments.
Reviewer 2 Report
There are no further comments.